# Linking Green Human Resource Practices and Environmental Economics Performance: The Role of Green Economic Organizational Culture and Green Psychological Climate

**DOI:** 10.3390/ijerph182010953

**Published:** 2021-10-18

**Authors:** Syed Mehmood Ali Shah, Yang Jiang, Hao Wu, Zahoor Ahmed, Irfan Ullah, Tomiwa Sunday Adebayo

**Affiliations:** 1Northeast Asian Research Center, Jilin University, Changchun 130012, China; mehmood.shah100@yahoo.com; 2School of Management and Economics, Beijing Institute of Technology, Beijing 100081, China; zahoorahmed83@yahoo.com (Z.A.); irfanullahkhan9214@gmail.com (I.U.); 3Department of Business Administration, Faculty of Management Sciences, ILMA University, Karachi 75190, Pakistan; 4Department of Business Administration, Faculty of Economic and Administrative Science, Cyprus International University, Nicosia, Northern Cyprus, TR-10, Mersin 99080, Turkey; twaikline@gmail.com; 5Department of Finance & Accounting, AKFA University, 1st Deadlock, 10th Kukcha Darvoza Street, Tashkent 100012, Uzbekistan

**Keywords:** human resource management (HRM), green psychological climate (GPC), green organizational culture (GOC), environmental concerns, economic performance

## Abstract

An eco-friendly environment with green strategies can help to achieve better environmental performance. However, literature on the relationship between green human resource management practices (GHRMP) and sustainable environmental efficiency (SEF) is limited. Moreover, there is limited knowledge about the factors that could mediate the relationship between GHRMP and SEF. Therefore, the present study examines the impact of green human resource management practices mediating through green psychological climate (GPC) and green organizational culture (GOC) for better environmental efficacy. For this purpose, the primary data on variables are collected by using structured assessment tools and analyzed through regression models. Unlike previous studies, this study adopts a mediation model and unfolds not only the role of green human resource practices in psychological climate and green organizational culture but also clarifies the mediating role of GPC and GOC in sustainable environmental efficiency. The findings unfolded that ecological factors such as green psychological climate, green organizational culture, and sustainable environmental efficiency are positively affected by green human resources management. In addition, green organizational culture and green psychological climate positively mediate the relationship between GHRMP and SEF. This study recommends adopting green human resource management strategies and increasing technical innovations to improve sustainability and economic performance.

## 1. Introduction

Human activities have increased global warming up to 1.0 degrees Celsius over pre-industrial levels. This global warming is leading to climate change, which in turn increases unexpected weather changes over time, such as devastating storms and droughts, heatwaves, and forest fires, ecological destruction, and environmental degradation. Thus, climate change is considered to be among the most challenging issues of the twenty-first century [1]. The significant impact of climate change on a local, regional, and international scale can be felt around the world, while the corporate sector is also at the forefront of all sustainability controversies [2]. Companies may also play an essential part in dealing with ecological problems [3] and today’s magic word is “environmental consciousness,” which has infiltrated every part of our lives and workplaces.

Late in life, our personal and professional habits started to affect the planet negatively, and we could not afford to ignore the implications. We need to change our living habits for better consequences. Undoubtedly, the corporate community participates in discussions on environmental issues and hence plays an essential role in resolving ecological threats. Green human resource practices is a concept that aids in creating a green workforce in a sustainable community that recognizes and values environmental stewardship.

As described by [4], green HRM is the application of HRM policies to encourage companies in utilizing their resources efficiently and promoting environmental stewardship, thus improving employee morale and happiness. Others describe green HRM as using HRM policies, principles, and practices to promote the productive usage of business resources while avoiding environmental harm caused by organizations [5]. Human resource management (HRM) is an essential administrative factor that deals with the most critical resource in organizations. Therefore, sustainability has become an important goal of HRM since environmental performance cannot be achieved without human efforts.

Moreover, [6] pointed out the degree of environmental commitment that determines the resulting set of environmental targets. Substantial work relevant to the manufacturing sector was performed in ecological improvement. A factor of 1.5 can be attained when pollution control is the strategic objective [5] while several scholars suggest a factor of four as a practical short-term eco-productivity objective [7]. Reduced environmental costs, above a multiplier of four, appear appropriate in terms of drastic changes coming from eco-innovation, and environmental load reductions up to a factor of 50 were listed for sustainability. The achievement of the fundamental global goal is heavily dependent on population growth, size estimations, supply, and demand dynamics [8]. In addition, we assert that the most critical part of development is green human capital management. In this context, analyzing the role of green practices for human resource management (HRM) in sustainable environmental management is the sole objective of this paper.

## 2. Literature Review and Hypotheses Development

An organization’s human resources department plays an essential role in developing the longevity of an enterprise [9]. Green management can be considered as to how an enterprise handles the climate with different techniques [10,11]. This term covers green procurement, green placement, green preparation, green success assessments, green rewards, and compensation. This activity must be effectively maintained. It is essential to learn how these healthcare institutions manage these activities in their fields. For hour managers, the critical issue in organizations will be working on numerous practices which improve their employees’ environmental performance and green behavior. A detailed analysis is needed to determine how green HRM activities are supposed and handled within these organizations in different practices. This specific report makes efforts to understand the connection between green human resource management strategies, environmental efficiency, and employees’ green behavior. 

Because of the growing global environmental deprivation confronting current and future generations, there is a stronger focus on environmental sustainability and sustainable development growth (SDG) over the last decade [12,13,14]. Indeed, the world was identified as the most critical market concern of the 1990s [15]. Since then, it has spawned the idea of “Go Green,” which is a worldwide buzzword among academics, the financial and environmental, and the public because of disturbing climate change and global warming [10,16,17,18,19]. This can be seen in the case of industries and other financial institutions where environmental issues affect living beings, such as global warming, acid rain, air and water pollution, ozone layer depletion, and climate change, and these are addressed through voluntary codes of conduct, including the United Nations Environment Programme-Finance Initiative (UNEP-FI), the Equator Principles for Project Finance, and others [20]. 

Environmental issues are perceived to be a very significant selling aspect that has led to the green marketing idea, which focuses on designing marketing campaigns that fulfill consumers’ environmental wishes. Like conventional marketing, greening is related to the concept of social marketing by identifying green practices and perceptions that could be used to develop green ideas and goods for encouraging the inclusion of fiscal, social, and environmental considerations in the implementation of values. Green initiatives may be considered a socially conscious policy that reaffirms an organization’s dedication to its corporate social obligations and acts as a benchmark to improve the efficiency of an enterprise. For most sectors worldwide, this is often deemed as an obstacle to incorporate adequate and acceptable green policies essential for modern enterprises. It is necessary to remember the environmental commitment to companies that is becoming an important consideration for consumers to support their decisions. 

Environmental performance evaluation (EPE) is a process developed by the International Standard Organization (ISO) to assist management in understanding and making decisions about an organization’s environmental performance by data collection and analysis and obtaining evaluation-based information on environmental performance. This study aims to provide the readers with basic understandings of green HRM, multiple valuable works on green HRM by others, and an improved range of green activities that can be used to build a healthy green workplace. The following are several attempts to suggest such HR environmental policies: to study the impact of green HRM practices on organizations’ environmental performance, mediating through the green climate and moderated through green organizational culture. Green HRM practices play a significant role in industry to boost environment-related issues. Organizations have enough chances of growth by being green and making a new responsive environment, which helps in substantial operational savings by decreasing their carbon impression.

By adopting green practices, firms can considerably reduce service costs by using energy-efficient, low-waste technologies. Green HRM can create a culture of concern for the well-being and health of our fellow employees.

Green HRM can improve the environment by recycling waste material, which makes the environment clean and sustainable. A sustainable environment enhances the air quality and ensures the health of the public. This study investigates the organization’s green HRM practices and finds ways to implement them to maintain the environment and improve business performance.

By adopting green practices, the environmental performance becomes sustainable, which helps the organization to achieve competitive advantages. A sustainable environment also makes society green, which makes sure the health safety of an individual. Green HRM practices in an organization and society ensure the sustainability of the domain.

### 2.1. Green Human Resource Management and Environmental Performance 

The concept of green management was initiated as a part of business strategy during the 1990s [21]. However, in the 2000s, it began to gain power. In essence, Wehrmeyer and Parker (1996) coined the term green HRM as the organized and intended integration of traditional HRM activities with the organization’s environmental priorities [22]. Moreover, [23] stressed the dire need to create correlations between human resource practices and sustainable development. In this context, a modern concept of economic growth is discussed that aims to promote politically, ecologically, and socially sustainable development through trade and foreign policy, financial and fiscal policies, agricultural, and industrial policies [24]. The literature on the degree to which HRM is green is frequently analyzed on a spectrum of traditional HR practices: employment research, recruiting and placement, induction, preparation, assessment of results, and awards [24].

Designing and implementation of new roles and positions to concentrate exclusively on environmental aspects of organizations can be referred to as reen work design [25]. It includes combining various roles, duties, and obligations in each work relevant to environmental protection. In other terms, the addition of environmental factors into the work definition and, at the same time, the integration of green competence in job specification is an intriguing part of green job-research [25].

Many businesses have already begun to adopt sustainability practices for environmental protection and each work description has started to directly provide at least one ecological maintenance requirement and even ecological concern [24]. The green recruiting and selection process is based on environmentally friendly methods for recruitment such as internet resources and reduced paper use during the recruitment phase, and the use of green attitudes at selection [26]. The green skills of individuals are essential to environmental success [27]. Therefore, it is worth remembering individuals who value sustainability practices and adopt simple ecological activities such as recycling, carpooling, and energy saving. On the other side, even applicants who respect ethical commitments will be drawn to environmentally friendly and “green employer” organizations [22]

Furthermore, [25] notes that green induction entails familiarizing new hires with the organization’s greening activities and empowering them to show green interpersonal citizenship. New workers must have a genuine understanding and strategy for their organizational environmental culture [28]. Organizations can take two approaches to this, i.e., general green induction and green work induction [29]. Under general green leadership, companies provide prospective members with basic knowledge on strategies and procedures on environmental sustainability. New hires are focused on sustainability projects unique to their work during specific eco induction. Today, all these tactics in companies prove their merit.

Environmental consciousness should be considered a precondition for achieving triumphant environmental success by corporate workers at all levels. Therefore, environmental education is essential to improve the mindset and actions of the organizational participants [29]. The focus was placed on providing such instruction for workers who would promote conservation and waste management practices. Training workers to generate green workspace research and energy conservation, execute rotations to prepare green management for the future, and improve green personal abilities may be regarded as valuable green training and growth strategies.

The green performance assessment is a performance assessment based on green standards that has a different aspect to the success in greening the performance feedback interview [30]. Such green activity will show its merit since once an action is assessed to evaluate a person, its perceived value increases, and efforts are made to adhere. Thus, using green measures in the performance assessment scheme will speed up employee adoption [17]. Moreover, [31] have welcomed the inclusion of environmental sustainability priorities in the performance assessment scheme, as it guarantees that workers’ progress is regularly provided with reviews.

This involves the adaption of programs to reward the development of renewable skills, the usage of monetary and non-monetary environmental incentives (such as benefits, cash premiums, sabbaticals, gifts), and connectint green pay and reward systems to the green system activities [5]. It must be emphasized that green habits are rewarded in the workplace, and carbon footprints are reduced. This can be seen as a possible instrument for promoting organizational environmental practices.

Green empowerment: encouraging corporate employees to make environmental choices and encourage them to take accountability for their actions that lead to cost appreciation, a sense of identity, and improved ecological efficiency. Such green behavior strengthens employees’ commitment to sustainability programs and their happiness after achieving their environmental targets.

People and their harmful activities have affected the world and the company [4,32]. Green human resources management functions can build an environmentally sustainable ecosystem for companies. The HR professionals and the cooperation of workers are responsible for protecting the environment [33]. Different activities in HRM may be used to render it environmentally friendly. So, according to Cherian and Jacob (2012), the organization will achieve good results if it integrates human resources activities with environmental policies. Furthermore, [31] indicated that if companies do not include their workers in ecological activity, they will affect their ecological success. However, [34] noted later that many companies now rely very actively on protecting their community by their workers.

Green recruiting and selection are regarded as the strongest HRM methods in presenting applicants with the green HRM plan. One of the challenges facing HR people these days is to recruit and attract skilled people. Many organizations are still struggling to maintain their climate. It might help us get candidates who take the climate and sustainability seriously. This would help companies hit their environmental sustainability goal. Moreover, [34] green training and advancement are often deemed an essential human resources strategy for organizational success in environmental performance. Environmental training is considered an essential method for developing human resources [33,35]. This type of training can create an employee ethos, such that green sustainable policies, including waste reduction and a pro-active attitude to the environment, are developed. According to the [31] training curriculum, such activities as environmental protection techniques, energy conservation, safety recycling, and waste management may be used. 

Green HRM practices can encourage people to engage in environmentally friendly behavior [36]. The relationship between green HRM practices, such as green employment, green recruitment, green choice, green education, and green rewards, and the positive environmental performance is up to date by many scholars [37]. With the help of green practices, organizations can achieve sustainability and competitive advantages. The essential elements of green HRM are sustainability and environmental activities, but a lack of resources in previous studies [37].

Business organizations showcase themselves as natural conservatives in a range to pull incredibly savvy experts with excellent green information, adopting green practices and maintainability issues (Rawashdeh and Karim Al-Adwan, 2012). (Cherian and Jacob, 2012) pointed out that organizations focused on greening human resource purposes may be more efficient and generate positive outcomes. Conversely, companies not associated with employees who engage in green activities may be less efficient in ecological performance. Furthermore, [31] suggested that ecological enterprises should focus on fascinating and selecting applicants with knowledge of the environment.

According to the author of [38], the impact of HRM practices on business execution can be seen in two ways. The central approach is a methodology guide that considers the effect of a series of standard HRM activities on the success of an organization. The second strategy offers unique insight by exploring the impact of such HRM operations on the company’s execution. Since environmental efficiency drives business success, businesses in different industries use strategic ecological improvement practices to achieve a competitive edge [39].

The economic and ecological success is highly dependent on its management policy [1]. Changes to the organization’s environmental performance policy, facilities, and staff responses to it are often part of the procedure. A significant boundary state is a transition and organizations address environmental concerns such as energy and water conservation through employee behavior improvement, greenhouse gas pollution reduction, increased recycling efforts, and increased usage of public transportation.

Employee involvement in environmental management initiatives is essential, and they are most involved in partnering with companies concerned about environmental concerns [40]. Using acceptable HRM procedures at all levels of the organizational phase, a review of companies with ISO 14001 qualification and their findings shows that organizations with high worker motivation have improved environmental efficiency. 

Some considerations are critical when implementing green HRM activities, such as selecting applicants with a solid experienced environmental understanding and providing staff with regular environmental training [30].

The efficiency of the organizational climate can be explained as activities that have a positive environmental effect. Organizations must carry out practical ecological programs to do this [41]. Both pieces of research indicated the diverse GHRM practices that have a favorable and essential effect on organizational environmental practices. This would also give the rivals a competitive edge. After the discussion mentioned below, hypotheses are developed:

**H1:** 
*GHRM has a positive and considerable impact on GPC.*


**H2:** 
*GOC is positively influenced by GHRM.*


#### 2.1.1. Green Psychological Climate

The green climate was identified as the climate for businesses that achieve sustainability objectives by implementing environmentally-friendly policies [42]. A green psychological environment is thus a consciousness of individual green policies, processes, and practices that represent the organization’s green values [5]. If the organization prepares a robust environmental policy, it signals the integrity of its employees at the core of the enterprise [33]. Through following green HRM policies, companies deliver messages to workers on their environmental concerns above and beyond merely economic incentives to include employees in green decisions and activities [31].

There is an association between the understanding of corporate policy and the productive green actions of employees through the green psychology of a corporate world (Norton et al., 2014). Employees cannot invest in the working climate as they are not directly liable for energy expenses and supplies [42]. It is, therefore, necessary to explain the green duty of the company. Regarding the proper conception and evaluation of jobs; adequate green award helps articulate green workplace responsibility, fosters employee understanding of green benefit, and encourages employee interest in green business. Green psychological climates mediate the connection between green HRM and environmental success in the current study.

**H3:** 
*GPC has a positive impact on SEF.*


**H4:** 
*GPC positively mediates between the GHRM and P and SEF.*


#### 2.1.2. Green Organizational Culture

The culture of an organization is supposed as “green” if the employees of the organization think and act outside the profit-seeking purposes to maximize the optimistic impact of organizational operations while at the same time reducing destructive operational events on the natural environment [43]. Organizations with a green culture tend to measure and change several policies to solve problems associated with the environment. Such an organization takes steps to integrate policies for environmental development in the organizational mission and vision (Afum, Agyabeng-Mensah, and Owusu, 2020). Having a committed green culture applies pressure and prompts manufacturers to stay faithful to corporate values.

As stated by [41], an organization desires to increase environmental performance which emphasizes the top management and takes a sensible struggle to invest in other organizational members concerning ecological initiatives. Such organizations integrate green practices into their mission statements to place corporate members and grow personnel skillful at solving environmental issues to achieve better quality environmental performance [44].

One of the main reasons for accepting a green culture policy is to confirm that the idea of environmental sustainability fills the thinking of all organizational employees. As organizations accept a green culture based on a persuasive strategy, where all corporate employees are fully involved, environmental performance will likely improve. Therefore, the current study will investigate the environmental performance of green HRM practices by moderating the role of green organizational culture. 

Previous research has shown that a green culture will benefit the workers of companies where they work, thereby influencing the association between green HRM and environmental change. To better explain the partnership, the present research examines green HRM activities utilizing ecological efficiency as an intervening element in the presence of a green organizational culture.

The sustainability program must be recognized as a company-wide goal that includes all aspects of business activity [45]. The corporate culture improves sustainability and gains competitive advantages. Specifically, organizational culture development embraces the sustainability issues that are a vital source of sustainable competitive advantage.

The definition of corporate culture also appeared in green literature. Green corporate culture is a collection of principles, icons, assumptions, and organizations, representing the commitment or willingness to be an environmentally sustainable organization. Organizational culture is characterized as “a set of common mental assumptions that direct the action and perception of an organization by identifying appropriate behavior for various circumstances [46]. Symbolism for environmental conservation and security inside a green society forms the attitudes and activities of association participants.

Many authors argued that green organizations must make drastic behavioral changes to address environmental problems. Greening aims to increase the quality of non-renewable and renewable resource use, minimize emissions, and modify organizations and procedures that perform operations in a reasonable way concerning the environment [47].

Businesses choose to follow a green culture approach as management respect and express concern in protecting the environment [48]. OGC is essential in helping businesses turn their renewable success approach into a green one. Management issues in industrial enterprises facing environmental constraints compromise two competing objectives. Through selecting the optimum level of green performance and competitive advantage, profits may be reduced and green performance to increase benefit [49]. Companies without green culture can need limited capital to invest in a green strategy. These services may be allocated to critical corporate needs rather than environmental regulations by its management.

**H5:** 
*GOC has a positive impact on SEF.*


**H6:** 
*GOC positively mediates between the GHRM and P and SEF.*


### 2.2. Model Development

Two hypotheses were used to describe this theoretical research model. In the first place, we used the ability-motivation-opportunity (AMO) theory. According to the AMO theory, capacity, motivation, and opportunity are the components in HRM activities that describe how HRM actions increase an organization’s human resources by increasing their competence, resulting in better outcomes, less waste, and a sustainable green climate. In addition, our study further supports the principle of the SVF, which confirms how personal beliefs affect employee behavior. Moreover, [42] has observed that personal environmental thoughts play a significant role in the eco-friendly actions of workers. Therefore, the SVF theory supports the structure suggested in this research by providing an organization with an atmosphere that promotes the personal beliefs of its workers and thus improves its environmental efficiency. In our study, the dependent variable is sustainable environmental efficiency (SEF). Because different multinationals may integrate the ethical responsibility and institutional push for green behaviors in their business operations with the use of environmental management practices to enhance their environmental performance. For this purpose, better results can only be achieved by focusing on sustainable environmental efficiencies and these have their impact on management systems. GHRM progressively investigates human resources and their environmental efficacy. The investigation of the relationship between GHRM and SEF can better explain the mechanisms through which they impact environmental and economic performances [50]. In contrast, the green human resources management (GHRM), green physiological climate (GPC), and green organizational culture (GOC) are used as a mediator between GHRM and SEF. Figure 1 shows the research model for determining the mediating impacts.

## 3. Methodology

### 3.1. Data Collection and Measurement

A quantitative study approach is chosen for the present study. Primary data is collected using questionnaire techniques and used for analysis by computing descriptive statistics and performing regression analysis to evaluate mediation and moderation effects within the variables. This study evaluated sustainability management strategies in human resources, environmental efficiency, and workers’ green behavior. The research focuses on the highest middle and lower category of staff in various organizations. Data is obtained by stratified random sampling, in which a sample size of 480 workers was taken from the highest, medium, and lower levels. The questionnaire was prepared for research staff to assess green HRM policies, environmental efficiency, and green conduct. The population of the study has consisted of Chinese firms. In this analysis, primary parameters are retrieved that reflect the Chinese marketing background and marketing activities. The details began with the respondents’ demographic profiles. The second section of the questionnaire consisted of ecological HRM activities, environmental and progressive behavior. Input on green HRM was provided on green preparation, green compensation and reimbursement, green performance assessment, and green promotion. Strongly agree = 5, agree = 4, Neutral = 3, Disagree = 2, Strongly disagreed = 1. All elements except demographics were assessed at the five-point scale. Table 1 describes the variables used in the study. Below Table 1 indicated variables with a description of items and the Likert Scale, which has been used for data collection.

### 3.2. Data Analysis and Measurement of Model

SMART PLS 3.3 (SmartPLS GmbH, Schleswig-Holstein, Germany) is used for data analysis to employ the PLS algorithms with bootstrapping to 5000 substitute samples [51] Smart PLS is one of the leading software applications for partial least squares structural equation modeling (PLS-SEM). PLS is an emerging multivariate data analysis method, making it easy for researchers, academics, or even journal editors to let inaccurate applications of PLS-SEM go unnoticed. It was established by [52] and the software has gained acknowledgment since its introduction in 2005. It is munificently accessible to academics and researchers and has a gracious and responsive user interface and enhanced reporting aspects. Different tests were used for data validity and reliability such as CFA, rho_A, data reliability, AVE, and CR. Various items were used to improve the reliability and validity of data computed by the 5-point Likert scale, such as 1 for strongly disagree to 5 strongly agree. The mostly 5-point Likert scale is used in research because less than five and more than 7-Likert scales are less accurate [51]. Table 2 presenting the demographic values of the data, and according to Table 3, all the values of CA, rho_A, CR, AVE, and FL is greater than the standard, e.g., the standard value for CA, rho_A, and CR is equivalent or superior to 0.7 and 0.5 for AVE [52]. Table 4 shows the discriminant validity of the model.

To get the final solution, the confirmatory factor analysis was completed on the data, the root mean square error of approximation (RMSEA), the standardized root mean square residual (SRMR) [51], normed fit indexed (NFI) [53] the squared Euclidean distance (d_ULS), the geodesic distance (d_G) values, among others, were all contained by the expected range, indicating that there was no considerable common method variance (CMV) among the data assembled. The anticipated structural model was then anticipated to assess path evaluation and complete model fit. According to the evaluation, demonstration data is fit and satisfactory. 

## 4. Results

As per findings the major fit indicators are Chi-square = 7526.023, SRMR = 0.043, NFI = 0.727, d_ULS = 0.913, d_G = 8.141 and RMS theta = 0.156. The compositions and the pathways statement a substantial portion of the variation in the endogenous constructs postulated. Table 5 is showing the R square values, Like, GOC = 0.678, GPC = 0.640 and SEF = 0.785. It means that model is significant and well explained, like 68%, 64% and 78%, respectively. 

Figure 2 shows the outer loadings and path analysis values from H1, H2, H3, and H5. Table 6 also contains beta, SD, T Statistics, and *p* Values. These values show that H1, H2, H3, and H5 have a strong positive relationship. According to the conclusions, the effect of “GHRMP” on “GPC” and GOC is (β = 0.800, *p* = 0.000) and (β = 0.823, *p* = 0.000), respectively. The impact of “GPC” on “SEF” is (β = 0.445, *p* = 0.000), and “GOC has the impact on “SEF” is (β = 0.520, *p* = 0.000).

Figure 3 and Table 7 are showing the mediation analysis of the model. According to results, “GPC” and “GOC” strongly mediates the between “GHRMP” and “SEF”. E.g.,: (β = 0.356, *p* = 0.000) and (β = 0.428, *p* = 0.000) respectively.

## 5. Discussions

This study has far-reaching implications for theory and practice. This GHRMP study adds to the persistent need for further research to bring together the various HRM outlets for achieving environmental sustainability. Environmental concerns are commonly regarded as a significant selling point. The goal of green marketing is to create marketing campaigns that satisfy consumers’ ecological desires. Green ventures can be seen as an economically responsible approach that reaffirms a company’s commitment to corporate social responsibility. This is often regarded as an impediment to the introduction and application of appropriate and suitable green policies in most industries worldwide. It is crucial to note that a company’s environmental responsibility is getting increasingly important as customers become more influential in their purchasing decisions.

Prospective employees are more apt to see GHRM-enabled companies as the place to work for their dreams, and they are more likely to show long-term environmental productivity. These results demonstrate GHRM’s potential to help organizations becoming more environmentally conscious. This knowledge would encourage physicians to effectively integrate GHRM into strategy, practice, and workforce training programs to attract qualified candidates. This has ramifications for organizational relationships, as organizations will emphasize the environmental friendliness of their hiring messages. Given that the report’s respondents were final-year students, businesses will likely benefit from open dialogue about their environmental commitments and achievements at various stages of the campus recruitment and selection process. Eco-conscious organizations would gain a strategic edge in recruiting technical workers and, as a result, win the fight for talent in this highly dynamic business environment if they had to make open and meaningful disclosures of their green practices. To build faith and show genuine concern for the environment, it is recommended that organizations back up their claims and accomplishments with facts. Pro-environmental information in employment advertisements has been identified as attractive, prestigious, and deliberate information because organizations that advocate environmental conservation are likely to positively impact potential recruits in the hiring process [54]. 

The present study also affects organizations that are not known for their sustainability activities. As a result, those organizations will either be overlooked or considered by potential employees while considering possible job opportunities. This study contributes to the limited academic literature on sustainable HRM literature and sustainable environmental efficiency. Previous literature does not analyze the mediating role of GPC and GOC in the nexus between GHRMP and SEF. By cultivating a positive corporate citizen role, the organizations that practice GHRM influence prospective workers’ choices to join and collaborate for them. Organizations known for their business citizenship are perceived as ideal places to work, and business citizenship increases their attractiveness. The increased appeal of these organizations has influenced people’s desire to work with them. As a result, this research established organizational attractiveness as an important psychological mechanism for understanding GHRM’s impact on prospective employees’ attitudes and affective reactions to the business. The studies of [50,54] blend a positive sustainability message with an ecological green community and climate somewhat support the findings of this study. By explaining the boundary restrictions of the GHRM/SEP relationship, the study has implications for the organizational selection process, which considers the intimate environment and green culture of prospective employees. To attract high-quality applicants, the corporation must hire environmentally conscious individuals who will support the organization’s GHRM efforts by proactively engaging in sustainability events and displaying green attitudes.

## 6. Conclusions

Human resource management (HRM) is an essential management factor that deals with one of the most valuable assets of a human resources organization. The whole HRM background is presently under consideration in the context of sustainability. By extending this statement, we argue that the essential part of sustainability is green human resource management. In this article, we concentrate specifically on green human resource management (GHRM), in which human resources management (HRM) manages an organization’s ecosystem. The study of [44] describes green HRM as using HRM policies to facilitate efficient resource use within companies and the cause of the ecosystem that further enhances employee productivity and satisfaction. Others characterize green HRM as using HRM policies, perceptions, and practices to support the sustainable utilization of business capital and prevent harm from environmental issues among organizations [55].

Organizations must understand the value of green human resource activities and their effect on environmental sustainability to compete in today’s business world. The primary goal of this paper was to investigate the impact of green human resource management activities on environmental performance. According to the association study findings, both environmental efficiency and green attitudes have a strong influence on green HRM activities. This study evaluated sustainability management strategies in human resources, environmental efficiency, and workers’ green behavior. The research focuses on the highest middle and lower category of staff in various organizations. Data is obtained by stratified random sampling, in which a sample size of 480 workers was taken from the highest, medium, and lower levels. All elements except demographics were assessed at the five-point scale.

Furthermore, the research sought to include comprehensive mediation and moderation models to emphasize the interaction more clearly. Because of the regression study, the organizational environment is a mediating variable. In contrast, green corporate culture serves as a moderating variable, reducing the influence of the contingent and independent variables. The findings prove that GHRMP and SEF relationship is positively mediated by GOC and GPC. 

### 6.1. Theoretical Implications

This paper highlights the emergence and dissemination of green HRM activities that positively support environmental performance. GHRM activities reflect a company’s vital assets as they are a sustainable competitive advantage. GHRM researchers aim to use the company’s resource-based perspective [56] to clarify the importance of GHRM activities in environmental efficiency and sustainability [57]. For example, [58]) argues that a green atmosphere or a green culture inside a company can be a source of sustainable, triumphant environmental success. Moreover, [59] argues that a GHRM framework can be a unique source of sustainable competitivity, particularly when its components have a high level of green experience both internally and externally. In short, most GHRM theories concentrate on the causal mechanism by which internal capital or processes such as green climate and culture may contribute to both corporate success, and environmental performance. Management fashion theory and institutional theory concentrate on the aspects of the outside world around organizations that affect organizations’ activities and resulting outcomes. Management is a comparatively transitory common belief spread by fashion designers that contributes to rational management success through a specific management technique. Management modes, including aesthetic fashion, are typically defined by quick, gloomy swings in management. 

### 6.2. Policy Recommendations

Due to the increasing amount of environmental concerns, organizations are under raising pressure to respond properly and for the purpose to implement sustainable business methods, and by following sustainable policies, the development of green human resource management (HRM) practices benefit both the organization and its stakeholders. It makes a strong vision to investigate how green HRM practices might improve environmental economic performance and its sustainable impact [60]. GHRM is an expansion of HRM and is a key player in developing policies, legislation and directing public awareness campaigns to educate employees about the importance of a green environment. GHRM policies and their practices are not limited to specific enterprises, it is for the whole area of any business and is responsible for green careers and green concerns in an environmentally friendly approach for better environmental and economically green performances [61]. Environmental protection through pollution deterrence, resource management, and lessened energy utilization provoke organizations to add green practices into their supply chain for green psychological climate, human resource strategies, and practices. It proposed the impact of green human resource management and green climate methods on effective market, social, financial, and environmental performance [62]. There is a dire need to uncover more effects of GHRM policies for employee satisfaction, economic performance, the link between GHRM policy practices and green management, reputation, quality, external and internal factors of organizations, employees interact with the organization, and most importantly, how these integrated GHRM policies are beneficial for better environmental performances [63]. Here, all the HRM activities, programs, and initiatives will be considered in terms of green psychological climate sustainability. An increment in these elements of green HRM results can help to achieve the organization’s green strategic goals. 

This research also explained how environmentally effective green HRM activities are required and how workers learn green behavior. These respondents should be used further about the studies’ direction to make up a more comprehensive survey. It may also be seen in other markets, such as the automotive and utility sectors. The current study recommends that GPC and GOC can be strengthened to achieve more environmental benefits of GHRMP in organizations. The present results urge the researchers to re-consider GHRM policies for more clarifications behind the present needs of organizations for developing a green environment. The significance of originality in promoting competitiveness, as well as the urgency of resource preservation, embracing environmentally driven technologies look to be a natural choice for promoting sustainable development for organizations and better economic performances.

## Figures and Tables

**Figure 1 ijerph-18-10953-f001:**
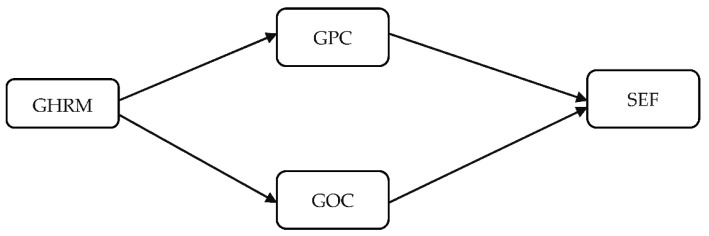
Conceptual model.

**Figure 2 ijerph-18-10953-f002:**
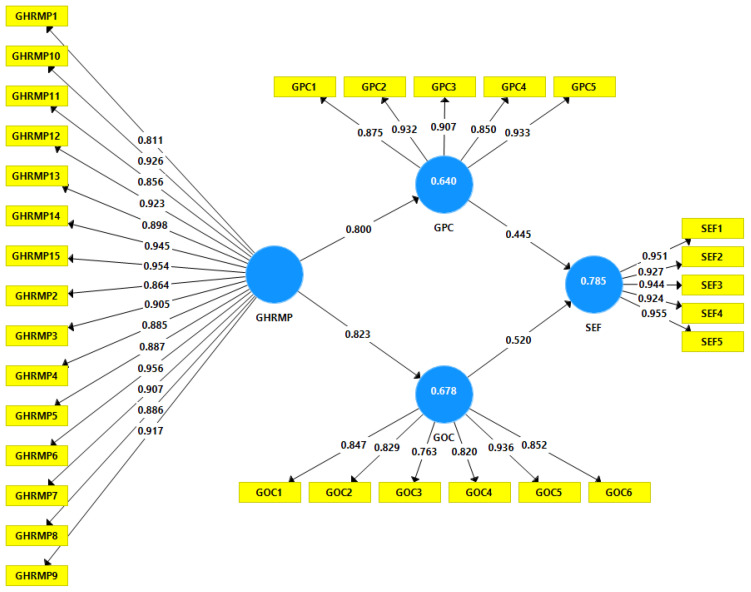
Factor loadings and direct path analysis.

**Figure 3 ijerph-18-10953-f003:**
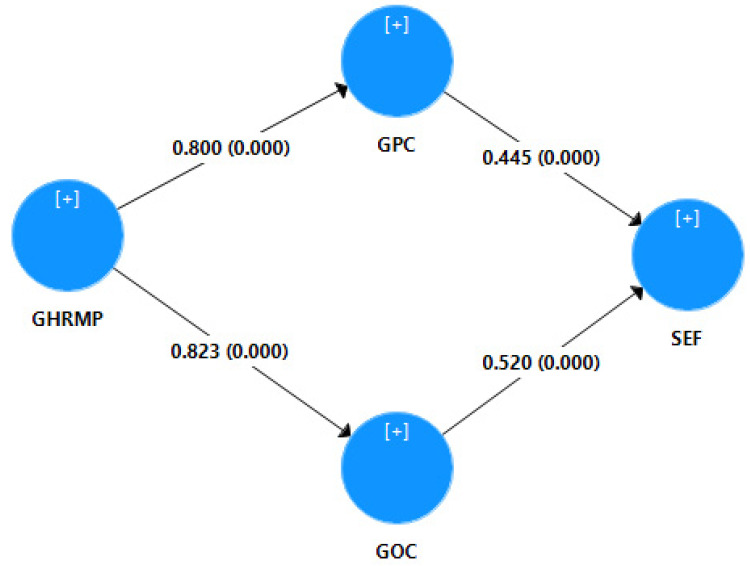
Mediator analysis.

**Table 1 ijerph-18-10953-t001:** Variables with the description used under the present study.

Variable	Description	5-Point Likert Scale
**GHRM**	It provides a modern concept to promote sustainable development by following innovative policies on environmental aspects of organizations and encourage people to engage in green behavior.	Strongly agree = 5, agree = 4, Neutral = 3, Disagree = 2, Strongly disagreed = 1
**GPC**	Providing green climate by implementing environmentally-friendly policies of organizations to explains the green duty and mediates the link between green HRM and environmental success for both employees and companies.	Strongly agree = 5, agree = 4, Neutral = 3, Disagree = 2, Strongly disagreed = 1
**GOC**	Organizations with green culture evaluate the diverse policies to solve environment-related problems and improve the environmental performance for better development and achieve the required target and vision.	Strongly agree = 5, agree = 4, Neutral = 3, Disagree = 2, Strongly disagreed = 1
**SEF**	It pushes green behaviors in organizations with the implementation of environmental management policies for enhanced performance.	Strongly agree = 5, agree = 4, Neutral = 3, Disagree = 2, Strongly disagreed = 1

**Table 2 ijerph-18-10953-t002:** Demographic information.

	Frequency	Percentage
** *Gender* **		
Male	254	53%
Female	226	47%
* **Age** *		
21–30 years old	198	41%
31–40 years old	185	39%
41–50 years old	97	20%
** *Education of Respondents* **		
Bachelors	240	50%
Masters	206	43%
PhD. Degree	34	7%
** *Type of Occupation* **		
Government Job	78	16%
Private Job	146	30%
Student	94	20%
Businessman	130	27%
Housewife	32	7%

**Table 3 ijerph-18-10953-t003:** Factor analysis, data validity, and reliability.

Constructs	FL	CA	rho_A	CR	AVE
** *GHRMP* **		** *0.984* **	** *0.984* **	** *0.985* **	** *0.814* **
GHRMP1	0.811				
GHRMP10	0.926				
GHRMP11	0.856				
GHRMP12	0.923				
GHRMP13	0.898				
GHRMP14	0.945				
GHRMP15	0.954				
GHRMP2	0.864				
GHRMP3	0.905				
GHRMP4	0.885				
GHRMP5	0.887				
GHRMP6	0.956				
GHRMP7	0.907				
GHRMP8	0.886				
GHRMP9	0.917				
** *GOC* **		** *0.917* **	** *0.925* **	** *0.936* **	** *0.710* **
GOC1	0.847				
GOC2	0.829				
GOC3	0.763				
GOC4	0.820				
GOC5	0.936				
GOC6	0.852				
** *GPC* **		** *0.941* **	** *0.950* **	** *0.955* **	** *0.810* **
GPC1	0.875				
GPC2	0.932				
GPC3	0.907				
GPC4	0.850				
GPC5	0.933				
** *SEF* **		** *0.967* **	** *0.968* **	** *0.974* **	** *0.884* **
SEF1	0.951				
SEF2	0.927				
SEF3	0.944				
SEF4	0.924				
SEF5	0.955				

**Table 4 ijerph-18-10953-t004:** Discriminant validity.

Constructs	GHRMP	GOC	GPC	SEF
**GHRMP**	0.902			
**GOC**	0.823	0.843		
**GPC**	0.800	0.688	0.900	
**SEF**	0.994	0.825	0.802	0.940

**Table 5 ijerph-18-10953-t005:** R Square Values.

	R Square
**GOC**	0.678
**GPC**	0.640
**SEF**	0.785

**Table 6 ijerph-18-10953-t006:** Direct path coefficient.

Hypotheses		β	SD	T Statistics	*p* Values
**H1**	**GHRMP -> GPC**	0.800	0.029	27.812	0.000
**H2**	**GHRMP -> GOC**	0.823	0.025	32.823	0.000
**H3**	**GPC -> SEF**	0.445	0.066	6.689	0.000
**H4**	**GOC -> SEF**	0.520	0.063	8.261	0.000

**Table 7 ijerph-18-10953-t007:** Mediator analysis.

Hypotheses		β	SD	T Statistics	*p* Values
**H5**	**GHRMP -> GPC -> SEF**	0.356	0.063	5.616	0.000
**H6**	**GHRMP -> GOC -> SEF**	0.428	0.062	6.871	0.000

## Data Availability

The datasets generated for this study are available on request to the corresponding author.

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
