# Peer review of "Linking Green Human Resource Practices and Environmental Economics Performance: The Role of Green Economic Organizational Culture and Green Psychological Climate"

_ijerph, 2021, doi:10.3390/ijerph182010953_

Round 1
Reviewer 1 Report
English and communication are still to be revised. A number of sentences are difficult to understand
The description of the questionnaire used is still rather unsatisfactory
The introduction is too lenghty and makes difficul to follow the reasoning behind the study
Author Response
Responses to Reviewers Comments
Manuscript Title:
Linking green human resource practices and environmental Economics performance: The role of green economic, organizational culture and green psychological climate
1st Reviewers’ comments and authors’ response
Thank you so much for your kind reviews. I/We tried our best to cover your concerned reviews to make our paper more attractive by following your suggestions.
Comment 1:
English and communication are still to be revised. Several sentences are difficult to understand.
Author’s response:
English language and the sentences that were difficult to understand have been improved.
Comment 2:
The description of the questionnaire used is still rather unsatisfactory.
Authors response:
The description of the questionnaire has been clearly described.
Comment 3:
The introduction is too lengthy and makes it difficult to follow the reasoning behind the study.
Authors’ Response:
An introduction is again reviewed and set according to the reviewer’s suggestion and the reasons behind the study have been mentioned in the last lines of the introduction
Reviewer 2 Report
This study examines that HRM practices which deal with environmental issues can enhance environmental performance. In a sense that HRM technics can be a basis for company-wise environmental performance, I acknowledge that this study has many things valuable. And I found that this revised manuscript presents a lot of improvements. However, some points I raised at the last round were not still under-developed. I suggest three additional works for publication.
First, in the introduction section, please succinctly state what is the main motivation of this study and why this is so important. The main objective of this study will be reflected as a dependent variable in the empirics. Given that the dependent variable is sustainable environmental efficiency, the authors may want to elaborate why SEF should be taken seriously and how this variable can capture the main goal of the study.
Second, I can understand what GHRM is. However, in the hypotheses as well as measurements, what aspects of GHRM are considered in this study should be clearly stated. For example, “H1: GHRM has a significant impact on GPC” cannot be a hypothesis because GHRM, as a system of practices, cannot be scaled as a variable. Does the concept capture whether a company has a GHRM practice, the number of GHRM practices, the scope of GHRM practices (which area GHRM covers, etc.), or whether employees perceive GHRM practices the company implement? The authors may want to specify the aspect of GHRM for their hypotheses. shou
Third, data collection and variable measurements should be further elaborated. In particular, the authors may want to specify how each variable is measured with what survey items, computation approaches, data sources, etc.
Author Response
2nd Reviewers’ comments and authors’ response
This study examines that HRM practices that deal with environmental issues can enhance environmental performance. In a sense that HRM technics can be a basis for company-wise environmental performance, I acknowledge that this study has many things valuable. And I found that this revised manuscript presents a lot of improvements. However, some points I raised at the last round were not still under-developed. I suggest three additional works for publication.
Thank you so much for your kind reviews. I/We tried our best to cover your concerned reviews to make our paper more attractive by following your suggestions.
Comment 1:
First, in the introduction section, please succinctly state what is the main motivation of this study and why this is so important. The main objective of this study will be reflected as a dependent variable in the empirics. Given that the dependent variable is sustainable environmental efficiency, the authors may want to elaborate on why SEF should be taken seriously and how this variable can capture the main goal of the study.
Authors’ Response:
The main reason and purpose of the study are mentioned. The dependent variable, sustainable environmental efficiency has been elaborated that captures the focus of the present study by following your suggestions.
Comment 2:
Second, I can understand what GHRM is. However, in the hypotheses as well as measurements, what aspects of GHRM are considered in this study should be clearly stated. For example, “H1: GHRM has a significant impact on GPC” cannot be a hypothesis because GHRM, as a system of practices, cannot be scaled as a variable. Does the concept capture whether a company has a GHRM practice, the number of GHRM practices, the scope of GHRM practices (which area GHRM covers, etc.), or whether employees perceive GHRM practices the company implements? The authors may want to specify the aspect of GHRM for their hypotheses.
Authors’ Response:
Variables are modified according to the model of study. The aspect of GHRM practices is mentioned here.
Comment 3:
Third, data collection and variable measurements should be further elaborated. In particular, the authors may want to specify how each variable is measured with what survey items, computation approaches, data sources, etc
Authors’ Response
Data collection and variable measurements are properly mentioned according to the terms and conditions of the proposed study.
Reviewer 3 Report
Many thanks to the authors for allowing me reading and commenting their work. I expect that my contributions help towards the improvement of the document quality.
The topic raised is very important, being at present in the agenda of most policy makers and International institutions worldwide.
That being said, there are some aspects I would like to highlight:
- In terms of the general overview of the article, there are some highlighted sections which deserve further attention in terms of formatting and content.
- In general, there is a need to deeply format the document, but the referencing does not follow the standards of the journal. Also the sections are not numbered.
- Professional proof reading is also required as many sentences are misleading. Some ideas are hards to follow.
- The abstract needs to be completely re-written as it does not provide the reader information about the purpose and the findings as well as its implications.
- The keywords also need revision.
- There is a need to clearly identify a second section devoted to the literature review, only after discussing the literature becomes possible to present a conceptual model - please rebuild to meet the expectable form. (consider removing philosophical model from the text - it is better to mention conceptual).
- Concerning hypothese 1 and 2 - "significant impact" is not sufficient it can either be positive or negative. The same for 3 and 4.
- There cannot be paragraphs without the accurate referencing (e.g. line 350 and following).
- Consider a table to encompass the variable description and its scales.
- Methodology descriptions as what can be found on lines 410-414 cannot appear in a paper.
- Table 2 needs formatting - it looks sloppy.
- Also from lines 434 to 446 the text does not add value from extant tables. It must provide conclusions to the reader, and critical thinking.
- Figure 2 needs to be simplified. Re-organized.
- There is a need to include a section devoted to the results (perhaps mix with the exsiting Discussion).
- I believe that some attention needs to be paid to the ecosystem - I think that the last section need to include a sub-section in regards to the policy making as well as the extant regulations and their influence over firms. I suggest a recent article related to eco-efficiency: Costa, J. Carrots or Sticks: Which Policies Matter the Most in Sustainable Resource Management? Resources 2021, 10, 12. https://doi.org/10.3390/resources10020012
- There is a need to contextualize the importance of the policy actions in shaping the organizational decisions as well as the human resorces behaviour towards such an important aspect like the green culture.
- Consider the other helices of the entrepreneurial ecosystem as further avenues of research as the adoption of these practices is part of a collective effort which cannot be easily separated in parts.
Best of luck with your research!
Author Response
3rd Reviewers’ comments and authors’ response
Many thanks to the authors for allowing me to read and comment on their work. I expect that my contributions help towards the improvement of the document quality.
The topic raised is very important, being at present on the agenda of most policymakers and international institutions worldwide.
That being said, there are some aspects I would like to highlight:
Authors response:
Thank you so much for your kind reviews. I/We tried our best to cover your concerned reviews to make our paper more attractive by following the terms and your suggestions.
Comment 1:
In terms of the general overview of the article, some highlighted sections deserve further attention in terms of formatting and content.
Authors’ Response:
Thank you so much for your response and your kind reviews. We have made a few changes according to your highlighted sections with terms and conditions by paying more attention.
Comment 2:
In general, there is a need to deeply format the document, but the referencing does not follow the standards of the journal. Also, the sections are not numbered.
Authors’ Response
As per the above discussion, sections are numbered and the whole document is deeply formatted. We are thankful for these useful suggestions, and we hope that you will kindly reconsider your decision.
Comment 3:
Professional proofreading is also required as many sentences are misleading. Some ideas are hards to follow.
Authors response:
We again proofread our paper according to your reviews and made our ideas clearer.
Comment 4:
The abstract needs to be completely re-written as it does not provide the reader information about the purpose and the findings as well as its implications.
Authors response:
The abstract is re-written about the purpose and suggestions of the present study are clearer now.
Comment 5:
The keywords also need revision.
Authors response:
Keywords were revised completely and are clearer now.
Comment 6:
There is a need to identify a second section devoted to the literature review, only after discussing the literature becomes possible to present a conceptual model - please rebuild to meet the expectable form. (consider removing the philosophical model from the text - it is better to mention conceptual).
Authors response:
The literature review section is acknowledged, and the conceptual model is rebuilt by following your suggestions.
Comment 7:
Concerning hypotheses 1 and 2 - "significant impact" is not sufficient it can either be positive or negative. The same for 3 and 4.
Authors response:
Thank you for your kind suggestion. All the Hypotheses are modified by following your recommendations.
Comment 8:
There cannot be paragraphs without accurate referencing (e.g. line 350 and following).
Authors response:
Thank you so much for your kind review. Line 350 and the following are with accurate references and are mentioned clearly.
Comment 9:
Consider a table to encompass the variable description and its scales.
Authors response:
Table with variable description has been added.
Comment 10:
Methodology descriptions as what can be found on lines 410-414 cannot appear in a paper.
Authors response:
These lines have been removed from the paper by considering your suggestion.
Comment 11:
Table 2 needs formatting - it looks sloppy.
Authors response:
The table is now numbered as three and it is set according to the followed methodology.
Comment 12:
Also from lines 434 to 446, the text does not add value from extant tables. It must provide conclusions to the reader, and critical thinking.
Authors response:
Lines 434 to 446 have been developed in methodology and described separately from the results section.
Comment 13:
Figure 2 needs to be simplified. Re-organized.
Authors response:
Figure 2 has been made simplified.
Comment 14:
There is a need to include a section devoted to the results (perhaps mix with the existing Discussion).
Authors response:
Results sections have been separated from discussions as per your suggestion.
Comment 15:
I believe that some attention needs to be paid to the ecosystem - I think that the last section needs to include a sub-section in regards to the policy-making as well as the extant regulations and their influence over firms. I suggest a recent article related to eco-efficiency: Costa, J. Carrots or Sticks: Which Policies Matter the Most in Sustainable Resource Management? Resources 2021, 10, 12. https://doi.org/10.3390/resources10020012
Authors response:
Thank you so much for your concern. The policy implementation section has been added by following your kind suggestions.
Comment 16:
There is a need to contextualize the importance of the policy actions in shaping the organizational decisions as well as the behavior of the human resource towards such an important aspect like the green culture.
Authors response:
Thank you for your consideration. Section with policy actions and human resource behavior towards green culture has been added.
Comment 17:
Consider the other helices of the entrepreneurial ecosystem as further avenues of research as the adoption of these practices is part of a collective effort that cannot be easily separated into parts.
Authors response:
The other helices of the entrepreneurial ecosystem for further research are mentioned.
Round 2
Reviewer 1 Report
The paper has been suffciently improved and now can be published
Author Response
Authors response:
Dear Reviewer,
Thank you so much for your response. Your comments about our paper are very precious to us. English language and style are fine/minor spell check has been corrected. Thanks.
Reviewer 3 Report
Many thanks for the opportunity of reading the improved version of your work.
I would like to ask the authors to re-read my previous comments and proceed accordingly as most of my suggestions were answered as being solved, however the document does not bring any novelty.
For instance, last sections fail to be commected to extant literature being broad and vague - add references and describe the precise context. Is the evidence pro or against the previous articles? Which are the poliy instruments.
Best of luck!
Author Response
Reviewer’s 3 comments:
I would like to ask the authors to re-read my previous comments and proceed accordingly as most of my suggestions were answered as being solved, however, the document does not bring any novelty.
For instance, the last sections fail to be connected to extant literature being broad and vague - add references and describe the precise context. Is the evidence pro or against the previous articles? Which are the policy instruments.
Best of luck
Authors response:
Dear Reviewer,
Thank you so much for your response. Your comments about our paper are very precious to us. We also want to make our paper more unique and innovative by following your suggestions. We carefully read your comments about the last section. We have made it more clear, broad, and precise by keeping your suggestions in my mind.
References have been added in the last section of the study with policy suggestions.
The novelty of the paper has been identified in the abstract as well as in the discussion part of the paper.
Evidence (pro or against) about literature has been added to the discussion. Also, we have clearly stated the novelty and argued that previous studies somewhat agree with the notion of this research; however, this research conducts a mediating analysis as well. So, it covers new dimensions.
This manuscript is a resubmission of an earlier submission. The following is a list of the peer review reports and author responses from that submission.
Round 1
Reviewer 1 Report
When the authors point out that the literature on green human resource management lacks knowledge, I suggest reviewing these phrases based on the authors , Jabbour, C.J.C.; Freitas, W.R.S.; Mishra, P.; Jackson, S.E.; Renwick, D.W.S.; between others.
The authors need to highlight the limitations of the research and the social, academic and empirical contributions clearly.
Based on the proposed conceptual model, the authors performed a structural equation modeling, therefore, I suggest seeing the works of the authors: Latan, H.; Hair Jr., J.F.; Henserler, J., Ringle, C.M.
It was also not clear how the questionnaire used was constructed and validated. I suggest seeing Synodinos, N.E.
The authors need to deepen the theoretical foundation, based on the constructs proposed in the research.
Reviewer 2 Report
The topic addressed is interesting and timely due: Green Human resource Practices. Unfortunately there are serious drawbacks that prevent me from expressing a positive assessment of the paper
The introduction is rather unnecessary lengthy and not well focused on the topic addressed. The literature review on GHRPis rather poorly explored. I would mention, for instance:
Saeed, B. B., Afsar, B., Hafeez, S., Khan, I., Tahir, M., & Afridi, M. A. (2019). Promoting employee's proenvironmental behavior through green human resource management practices. Corporate Social Responsibility and Environmental Management, 26(2), 424-438.
Yong, J. Y., & Mohd-Yusoff, Y. (2016). Studying the influence of strategic human resource competencies on the adoption of green human resource management practices. Industrial and commercial training.
Renwick, D. W., Redman, T., & Maguire, S. (2013). Green human resource management: A review and research agenda. International journal of management reviews, 15(1), 1-14.
The hypotheses are not clearly stated; they are rather presented in such general way that makes impossible to the reader to understand what is really the paper about. It is impossible understand why are the specific aims of the paper
authors do not offer information on the development of the questionnaire they administered, What they did not use already validated scales? How the measures they used are pertinent to the hypotheses? All this generates great confusion and difficulty in understanding the real core of the study.
The conclusion and the discussion suffer from being very generic and iit is impossible to understand how they are really grounded on the data offered, due to the previously mentioned problems.
Reviewer 3 Report
This study examines that HRM practices which deal with environmental issues can enhance environmental performance. In a sense that HRM technics can be a basis for company-wise environmental performance, I acknowledge that this study has many things valuable. However, there are some concerns which should be clearly and succinctly addressed. Let me discuss the points for publication.
- Tautological arguments
First, the argument might be tautological. Green HRM is designed to enhance environmental performance. Thereupon, whether a given firm implements practices of green HRM can be an outcome as well as a cause for environmental performance. It would be more interesting if unexpected consequences take place depending on how the given firm implement green HRM practices. In other words, even though firms employ green HRM practices ceremonially, the actual implementation would not be perfectly matched to the original goals for the green HRM – so called green washing. I think how green HRM practices are (divergently) implemented across firms is more valuable to the audience in the environmental management area.
If the authors intended to illuminate the roles of HRM practices, which affect particular green behaviors of employees in a company, the mechanism of green HRM practices toward environmental economic performance should be elaborated. For example, how AMO and SVF theories can be applied to elaborate the main arguments of this study should be clearly stated. Otherwise, readers may not be able to understand and interpret the empirical results.
Further, the outcome variable (or dependent variable) is not clear. Environmental economics performance and environmental efficiency are interchangeably used. The authors may want to clearly define the focal variables, which is critical for publication.
- Perception vs. behavior & individual vs. organization
The authors may want to reconsider the survey method to capture the focal variables. The survey items seem unfounded, and some items don’t precisely capture the concept. More importantly, the research design should discern perception and behavior. Also, this study heavily depends on the self-reports of the respondents regarding firm behaviors. A person’s perception may not be representative to the entire organization. The authors may want to clarify how the measures can precisely capture the theoretical concepts.
- Survey items
Related to the main argument, the measures should be revisited. The correlation between environmental performance and green corporate culture is more than 0.9, which indicates that these terms are not distinctive. As discussed, some of the survey items seem irrelevant to the main concepts. More elaboration is needed.
- Mediation and moderation
The research design, as well as hypotheses, on the moderating and mediating effects should be restated or even reconsidered. How the meditation or moderation effects can take place should be elaborated and the empirics should be refined. If the authors ran a structural equation model, how likely the models are fitted to the main arguments should be explained and also the authors should provide the proper interpretations on the results. Otherwise, readers may have difficulties in understanding the findings of this study.
Hope these comments help.